# Creating a Scoring System with an Armband Wearable Device for Table Tennis Forehand Loop Training: Combined Use of the Principal Component Analysis and Artificial Neural Network

**DOI:** 10.3390/s21113870

**Published:** 2021-06-03

**Authors:** Wen-Lan Wu, Jing-Min Liang, Chien-Fei Chen, Kuei-Lan Tsai, Nian-Shing Chen, Kuo-Chin Lin, Ing-Jer Huang

**Affiliations:** 1Department of Sports Medicine, College of Medicine, Kaohsiung Medical University, Kaohsiung 80708, Taiwan; taiga1115@gmail.com (J.-M.L.); koenjfy@gmail.com (C.-F.C.); 2Department of Medical Research, Kaohsiung Medical University Hospital, Kaohsiung 80708, Taiwan; 3The Master Program of Long-Term Care in Aging, College of Nursing, Kaohsiung Medical University, Kaohsiung 80708, Taiwan; 4Ph.D. Program in Biomedical Engineering, College of Medicine, Kaohsiung Medical University, Kaohsiung 80708, Taiwan; 5Physical Education Center, Kaohsiung Medical University, Kaohsiung 80708, Taiwan; tkl@kmu.edu.tw; 6Department of Applied Foreign Languages, National Yunlin University of Science & Technology, Yunlin 64002, Taiwan; nianshing@gmail.com; 7Center for Physical and Health Education, Si Wan College, National Sun Yat-sen University, Kaohsiung 80424, Taiwan; linkuochin@mail.nsysu.edu.tw; 8Department of Computer Science and Engineering, National Sun Yat-sen University, Kaohsiung 80424, Taiwan; ijhuang@cse.nsysu.edu.tw

**Keywords:** wearable technology, electromyography, inertial measurement unit, artificial neural network, table tennis teaching

## Abstract

Background: This study presents an intelligent table tennis e-training system based on a neural network (NN) model that recognizes data from sensors built into an armband device, with the component values (performances scores) estimated through principal component analysis (PCA). Methods: Six expert male table tennis players on the National Youth Team (mean age 17.8 ± 1.2 years) and seven novice male players (mean age 20.5 ± 1.5 years) with less than 1 year of experience were recruited into the study. Three-axis peak forearm angular velocity, acceleration, and eight-channel integrated electromyographic data were used to classify both player level and stroke phase. Data were preprocessed through PCA extraction from forehand loop signals. The model was trained using 160 datasets from five experts and five novices and validated using 48 new datasets from one expert and two novices. Results: The overall model’s recognition accuracy was 89.84%, and its prediction accuracies for testing and new data were 93.75% and 85.42%, respectively. Principal components corresponding to the skills “explosive force of the forearm” and “wrist muscle control” were extracted, and their factor scores were standardized (0–100) to score the skills of the players. Assessment results indicated that expert scores generally fell between 60 and 100, whereas novice scores were less than 70. Conclusion: The developed system can provide useful information to quantify expert-novice differences in fore-hand loop skills.

## 1. Introduction

As wearable devices become cheaper and more popular, athletes have gradually begun to use them to analyze the extent and quality of their training. Apps installed on wearable devices can provide personalized fitness data and assist in self-tracking and quantifying exercise data [1,2,3]. In addition to benefiting professional athletes during training, wearable devices can assist physical education teachers. Monitoring systems can improve the effectiveness of students’ learning and allow them to practice sports-related skills more easily. One wearable device, the Myo Armband (Thalmic Labs, Inc., Kitchener, ON, Canada), was used in several applications, such as gesture recognition [3,4,5], gesture based controller [6,7,8], or providing medical solutions [9,10]. In the field of sports, Raina et al. (2017) used the Myo Armband to track changes in muscle exertion and arm swing as badminton shots were performed [3]. The advantage of this system is its sensors, namely, eight dry-surface electromyography (EMG) sensors and one nine-axis inertial measurement unit (IMU) sensor (three-axis gyroscope, three-axis accelerometer, and three-axis magnetometer), which are built into a simple, lightweight elastic armband.

Studies have used various sensors and data analysis methods for table tennis observations, such as video analysis to track the ball’s trajectory during service [11]; force sensors and electronic circuits to monitor the net [12,13] and detect ball–net impact during service [13]; inertial sensors mounted on the paddle to assess the ball’s speed and spin [14], estimate the trajectory of the paddle [15], and detect the type of stroke [16,17]; and inertial sensors worn on the limbs to detect shots [18], estimate kinematic parameters [19], and recognize stroke motions [20,21,22,23,24,25,26]. Machine learning classification algorithms are a reliable approach for recognizing basic human motions. The authors of related studies have recommended various expert machine learning models for table tennis observations, such as the k-nearest neighbor algorithm [20,21], support-vector machines (SVMs) [13,14,16,17,21,24], neural networks [17,18,22,23,25], or Long Short-Term Memory deep learning methods [26], all of which can achieve a sufficient level of accuracy for recognizing and classifying table tennis strokes. Because of their excellent nonlinear mapping and learning capabilities, neural networks can fully link information into network nodes and create a network mod-el with which to produce an effective prediction model. The aim of this study was to use artificial neural networks (ANNs) to build an expert model of a proposed system.

In addition to kinematic data, physiological signals, such as EMG signals, can be used to detect the degree of muscle activation during any movement [27]. In a table tennis study, Kondrič et al. (2006) examined the difference in upper arm EMG signals between forehand top spin strokes when 38-mm and 40-mm balls were struck [28]. Meghdadi et al. (2019) used EMG analysis to investigate differences among players with and without shoulder impingement syndrome in timing and activity intensity in the shoulder girdle muscles when a forehand topspin loop was being performed [29]. Chanavirut et al. (2017) assessed the effects of 6 weeks of training on wrist muscle strength, grip strength, and performance and used EMG as a measure of amateur players’ wrist flexor and extensor muscle conditions [30]. These studies suggest that EMG signals can be used to analyze players’ performance by indicating muscle activation during play. For this reason, the aim of this study was to use EMG signals generated by the wearable Myo Armband to under-stand differences in forearm and wrist movements among novice and expert table tennis players and to design an expert system for providing feedback on training.

Although wearable artificial intelligence devices can help to classify table tennis strokes [16,17,26] and distinguish the players’ levels [26], the basic classifications of “expert” and “novice” do not give coaches sufficient information to support their teaching. To bolster table tennis education, this study proposed an intelligent expert system based on principal component analysis (PCA) that projects data points from IMU and EMG sensors built into the Myo Armband onto each principal component. The information derived from this data can be interpreted into appropriate instructions for teaching. In the system, data from IMU and EMG signals during each stroke are extracted and clustered. After PCA is used to filter out the input variables of machine learning, the cluster results reveal the corresponding skill requirements. Using this innovative combination of ANN and PCA approaches for the development of our system, it can provide the automated scoring checks for each player.

## 2. Materials and Methods

### 2.1. Participants

The participants in this study were six expert male table tennis players on the National Youth Team (mean age 17.8 ± 1.2 years) and seven novice male players (mean age 20.5 ± 1.5 years) with less than 1 year of experience. Players who in the 6 months prior to the study had had musculoskeletal disorders or obvious trauma to the upper or lower extremities (such as fractures) that resulted in pain or inability to perform the test normally were excluded from the study. Each participant was informed of the purpose and content of the experiment and signed a consent form beforehand. The study was conducted in accordance with the Declaration of Helsinki, and the protocol was approved by the Ethics Committee of Kaohsiung Medical University Chung-Ho Memorial Hospital (KMUHIRB-E(I)-20190192).

### 2.2. Experimental Instruments and Equipment

The Myo Armband (Thalmic Labs Inc., San Francisco, CA, USA, Figure 1) is a wearable armband for the forearm with eight parts. Each part consists of an EMG sensor containing three metal electrode contacts and a signal amplifier. The device also features a nine-axis IMU unit, installed in the fourth sensor (Figure 1); it contains a three-axis gyroscope, a three-axis accelerometer, and a three-axis magnetometer. The dynamic ranges of measurement for accelerometer and gyroscope were ±16 g and ±2000 deg/s, respectively. The armband provides two sources of information: EMG signals with a frequency of 200 Hz and IMU signals with a frequency of 50 Hz [31,32]. Wireless data transmission occurs through a Bluetooth 4.0 connector. Previous study has reported that the armband sensor data is comparable to conventional EMG and IMU data and has high test-retest (trial-to-trial and day-to-day) reliability of acceleration, gyroscope, and EMG data [33]. The armband is light weight (93 g) and has an adjustable circumference ranging from 19 to 34 cm; thus, wearer comfort was considered in its design. It was worn on the dominant arm during data recording. Sensor 4 (IMU unit) was positioned approximately 3 cm below the lateral epicondyle, which is the origin of the extensor tendons of the wrist and fingers, and oriented in the direction of the *X*-axis (negative direction) toward the palm. The armband was then fastened accordingly. According to the previous study [34], in above-mentioned armband setting, the EMG signals in sensors 3, 4, 5 correspond to the activation of the wrist extensors. In contrast, sensors 1, 7 and 8 correspond to the wrist flexors, and the signals in sensors 2 and 6 correspond to both of them.

### 2.3. Data Acquisition and Processing

The SMARTPONG table tennis robot (TJ-3000, TEH-JOU Science and Technology Co., Ltd., New Taipei City, Taiwan) was used to serve backspin balls at a rate of 30 balls per minute. Each participant had one minute (30 balls) forehand loops practice to familiarize the ball speed, then performed 10 forehand loops, and the data on 8 of these loops were analyzed. The data acquisition was stopped and retested according to the actual situation such as participant failed to keep up with the serve speed. Data from each stroke were then divided into backswing (BP) and forward swing phases (FP) according to the forearm angular velocities measured by the *z*-axis gyroscope (Figure 2). For the analysis of phases, the following steps were taken. First, we defined the maximal positive value of the forearm angular velocity as point A. Second, we identified point B as the point where the angular velocity returned to 0 before point A. Third, we identified the point where the peak negative angular velocity before point B occurred and then found the point at which velocity that returned to 0, point C. Fourth, we identified point D, a drop to 0 after point A. The intervals between points C and B and points B and D represent the BP and FP phases, respectively. We defined the greatest positive and negative angular velocities as the maximum forearm angular velocities in the FP and BP phases, respectively. The peak acceleration values were also recorded, and they could be either positive or negative values. In addition, data from the eight EMG sensors were segmented according to the previously mentioned intervals. Signal rectification and envelope detection with cutoff frequencies of 10 Hz were used to process the data from the EMG signals to yield a smoother output. The final step was to calculate the integrated EMG (iEMG) values. In total, 14 parameters were recorded in each phase: the three-axis peak forearm angular velocity, the three-axis peak forearm acceleration, and the eight iEMG values. Data after each capture were obtained, and the subsequent stroke data were retrieved from the remaining data to avoid repeated capture of the same data. Capture ended after data from eight strokes were recorded. All analyses and machine learning procedures in this study were performed in Python (version 3.7.8) using the Jupyter Notebook. The research data were standardized such that they had zero mean before modeling.

### 2.4. Feature Selection and Modeling

Figure 3 shows the steps of the modelling process. This study used PCA when selecting features to reduce the input space in the ANN models. PCA selects variables according to the magnitude (from largest to smallest absolute value) of their coefficients (loadings). Kaiser criteria, according to which principal components with eigenvalues greater than 1 are retained for further analysis, were used to select the number of components retained in the PCA. After the principal components were extracted, varimax rotation was performed to achieve the linearly independent factors and create a straightforward, easy-to-interpret structure. Finally, after the dimension reduction, the factor score data were used as the neural network input. The factor score coefficient matrix was recorded to calculate the factor scores of the new input data outside the model. A total of 160 pieces of data (2 levels [5 expert + 5 novice] × 2 phases × 8 strokes) were included in the neural network models, with 80% used for training and 20% used for testing the models. Data for the remaining three participants (one expert and two novices) were used as the new data set for testing. We focused on performing multiclass classification on the players’ levels (expert and novice) and motion phases (BP and FP). Table 1 shows the label names for each player action. To create a classification model, a deep learning neural network (see Figure 4) was used with three hidden layers, each containing 20 neurons, and the rectified linear unit activation function (ReLU) was applied. The last layer was a sigmoid output layer with four nodes, one for each class.

### 2.5. Model Evaluation

We calculated the accuracy, recall, precision, and F1-score from the confusion matrix to fully evaluate the model. The accuracy is the ratio of correct predictions to the total number of predictions. Recall is the ratio of true positive predictions to all positive predictions. In contrast, precision measures the number of correct positive predictions, which is the ratio of true positives to total positive predictions. F1 score is the weighted average of precision and recall. For good classifiers, accuracy, recall, precision, and F1 score should be nearer to 100%.

### 2.6. Table Tennis Skill Scoring System

For automatic scoring, the system used a standard deviation scaling method to transform the PCA factor scores on a scale of 0 to 100. First, the average of the original factor scores for all experts in the model was set to 80 points. One standard deviation away from the mean was 70–90 points and two standard deviations from the mean was 60–100 points; a linear regression model was calculated to rescale and transform the raw data based on a scale of 0 to 100 points for each phase. Next, the system transformed the factor scores of the new input data according to this scale and displayed the results on a radar chart to visualize performance.

## 3. Results

Table 2 shows the mean (± SD) values of the 14 features across the two player levels during the BP and FP phases. The peak acceleration or angular velocity can be positive or negative. Positive or negative values represent different directions of movement. The value of the *z*-axis gyroscope is much larger than that of the other two axes, meaning that *z*-axis is the main rotation axis of the forehand loop drive; therefore, we used this axis for distinguishing between the phases.

Table 3 shows the significant factors extracted by the PCA. The Kaiser–Meyer–Olkin Measure of Sampling Adequacy value for the factor analysis was 0.896. An ideal score is one above 0.8, indicating that the factor analysis was useful for our data. In addition, the value for Bartlett’s test of sphericity was 3589.490 (degrees of freedom = 91, significance (Sig.) < 0.01). This test measures whether a correlation matrix is significantly different from an identity matrix. Our Bartlett’s test value was significant (Sig. < 0.01), indicating that our variables had a high covariance and that a strong relationship existed among them. This result meant that the correlation matrix was suitable for factor analysis. Only the first two principal components met the criteria of having an eigenvalue greater than 1 and were kept for further analysis. The cumulative percent variance was as high as 83.503%. Figure 5 displays the score plot of the first two principal components. These scores are labeled as PC1 (explosive force of the forearm) and PC2 (wrist muscle control). On the plot, scores falling close together have similar profiles, whereas those far apart are dissimilar. In Figure 5, data on the expert players’ FP phase fall close together in the upper right-hand corner, representing similarities in performance; all of the experts had faster arm swing and a higher degree of muscle control. Their data can be clearly distinguished from those of the novices. The data for backward swing fall on diagonally opposite quadrants of the plot. The difference between experts and novices in backward swing was small and less easy to distinguish.

Overall, the model’s recognition accuracy was 89.84%. As Table 4 shows, the prediction accuracy was 93.75%, and the macro average recall, precision, and *F*_1_-score were 0.93 for the model’s testing data set. For the new testing data set (not used in the model-building process), several measurements slightly decreased: accuracy dropped to 85.42%, and the macro average recall, precision, and *F*_1_-score were 0.86, 0.85, and 0.84, respectively. The new testing dataset was fully independent from the training dataset and was more representative of the general population; it is not a subset sampled for training. A true independent validation set can be used more reasonably to evaluate the model performance in the real environment. Although this approach decreased the accuracy slightly, it remained within an acceptable range.

Figure 6 presents radar charts of the average performance for the one expert and two novices comprising the new testing data set. The system created a linear regression model to transform the factor scores into a 0 to 100–point scale. The expert’s four average scores fell between 60 and 100, while the novices’ scores generally fell below 70. The standard deviation of each person’s scores in the four items presented in Figure 6 indicated that it falls within a reasonable range, proving that the system had satisfactory reproducibility in the judgment of personal skills. Furthermore, the experts had smaller standard deviation, meaning that their movements were more stable.

Differences in the two novices’ skills can also be identified. For example, Figure 6 shows that in the FP phase, the PC1 score was lower for new novice 1, leading us to infer that new novice 1 had sufficient wrist stability (PC2) for paddle control but lacked attack power (PC1). By contrast, the PC2 score for new novice 2 was lower in the FP phase, meaning that his main problems were a lack of grip strength and wrist stability for paddle control in the FP phase. The results of the experiment show that the scoring system created in this study can effectively detect an individual’s errors.

## 4. Discussion

In a study by Wang et al. (2019), three portable six-degree-of-freedom inertial sensors were placed on the upper arm, forearm, and wrist. The sensors detected stroke motions and recognized technical movements during table tennis training [20]. After PCA dimension reduction, an SVM algorithm was able to recognize and classify strokes. The authors stated that the main advantage of PCA is that the principal components became orthogonal to each other after the dimension reduction, which can eliminate interaction among components in the original data. We found that angular velocity and acceleration were highly correlated, and we classified them into the same component (PC1) after PCA. To render the data easier to interpret after the dimension reduction, we defined the principal component as the explosive force of the forearm (PC1). Angular velocity and acceleration increased significantly with greater explosive force. The data obtained by integrating the measures of acceleration and angular velocity on the forearm provided information about the trajectory and orientation of the arm players used to execute strokes. In addition, the trained classification model used in the study by Wang et al. reached a 96% recognition rate for five common strokes. The recognition rate of the two phases in our study using the ANN model was nearly 100% both for the testing data set and the new testing data set. Therefore, our model has clear practical importance and has broad usage potential. In addition, the multiclass classification rate (2 phases × 2 levels) was approximately 90%; occasional classification errors occurred when distinguishing between novices and experts. The sample characteristic scatter plot shows that movements, especially in the BP phase, were difficult to distinguish between novice and expert; the difference between the experts and novices was small. Experts had slightly stronger muscle recruitment (the scatter diagram shows an upward tendency on the *Y*-axis), but this difference was not remarkably large. To accurately detect whether a player has performed the BP motion correctly, that is, using the body as an axis to turn the waist back and extending the distance between the hand holding the paddle and the incoming ball, researchers of future studies should attach IMU sensors to participants’ waists, facilitating the collection of more accurate and detailed information.

After PCA, the second component, PC2 (wrist muscle control) was extracted. Contrary to popular belief, table tennis is a whole-body sport. The limitation of this study was that it only focused on the differences in minute adjustments of the forearm and wrist between experts and novices and did not analyze whole-body movements, such as the contribution of upper arm [28,29] and lower limb muscles [35,36,37]. Moreover, players’ fingers and wrists are integral parts of performance from a young age. In the entire movement that involves making contact with the ball, the fingers and wrist play a critical role in controlling position, arc, and spin. For example, the paddle must not be clenched too tightly or else the muscles will easily be fatigued and unable to release a burst of strength upon contact with the ball. The paddle must also not be held too loosely or else the orientation of the paddle will shift, and the player will not be able to concentrate their strength into one area of the paddle. Further investigations can improve our understanding of the benefits of finger and wrist strength during training.

Hegazy et al. (2020) addressed the use of infrared depth cameras on online matches to detect and classify players’ stroke errors in incorrectly played points [38]. Their results showed an average accuracy of 88–100% for backhand push and forehand push. In physical education courses, the most common strokes that beginners learn are the backhand push and the forehand push. However, the forehand loop is likely the most crucial stroke advanced table tennis players learn. The forehand loop builds on the foundation of the forehand drive by adding additional speed, spin, and power. A player should have mastered the basic strokes and developed a technically correct forehand drive before beginning to learn the forehand loop. The motion of the forehand loop resembles an exaggerated forehand drive, but with a lower stance, slightly more weight transfer, slightly more rotation, slight closure of the bat angle, faster acceleration, and more brushing of the ball. A player should always strive to improve their loop by adding more speed or spin. When returning a ball with spin, players must exert greater muscular activity when receiving a backspin forehand drive than when receiving a topspin forehand drive [39]. From the perspective of learning concepts, through appropriate integration of technology in physical education, such as real-time personal video records combined with expert videos, and integrating knowledge education, such as automatically displaying tips to improve stroke quality, beginners can acquire skills accurately and more directly after receiving feedback from our expert system.

One limitation of this study was the small sample size, which is a problem that most cost- and time-intensive data collection experiments must overcome in modeling. A small sample size can lead to biased machine learning performance estimations. First, this study used train/test split approaches to produce robust and unbiased performance estimates regardless of sample size. This ensured that the data used to validate the classifier were not among the data used for training, thus avoiding polling training and testing data to produce an almost unbiased estimate. In addition, the higher the ratio of features to sample size, the more likely the machine learning model is to fit the noise in the data instead of the underlying pattern. This study used PCA to reduce the number of features used to develop the model, while reducing the feature-to-sample ratio, to eliminate inherent noise and overcome the bias caused by the data dimension. Lastly, even though we did not perform feature selection on new testing data for data modeling, the prediction accuracy of the new testing data was still maintained at 85.42%. This indicates that although the number of modeling samples was not large, the established model should have controlled the problem of overfitting, and it is suitable for new data sets. Furthermore, because only men were included for modelling, including gender as a feature in the future will enable generalized observations of different gender groups. In machine learning, data satisfying a normal distribution are beneficial for model building. This is because many mathematical models for machine learning are explicitly calculated from the assumption that the distribution is bivariate or multivariate normal. In the present study, after further analysis, the quantifying outlyingness of the data in the multivariate domain with the Mahalanobis distance estimation and chi-squared test was satisfactory. Only 2 out of 160 multivariate outliers (and a possible violation of multivariate normality) were found, suggesting that our data fit a multivariate normal distribution.

## 5. Conclusions

The e-training system developed in this study can classify periods of table tennis strokes with high accuracy and can provide accurate scores to quantify player performance in each period. The results of the study suggest that our automatic assessment system can provide useful information for developing a table tennis training support system.

## Figures and Tables

**Figure 1 sensors-21-03870-f001:**
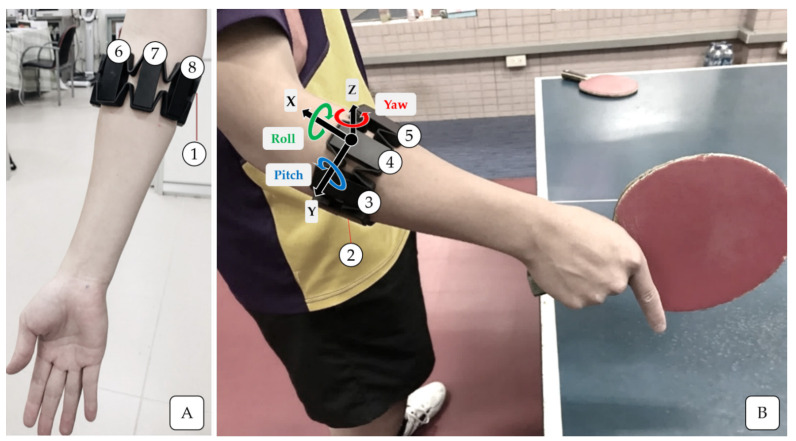
Placement of the Myo Armband on the forearm (**A**) and the direction of IMU sensors (**B**).

**Figure 2 sensors-21-03870-f002:**
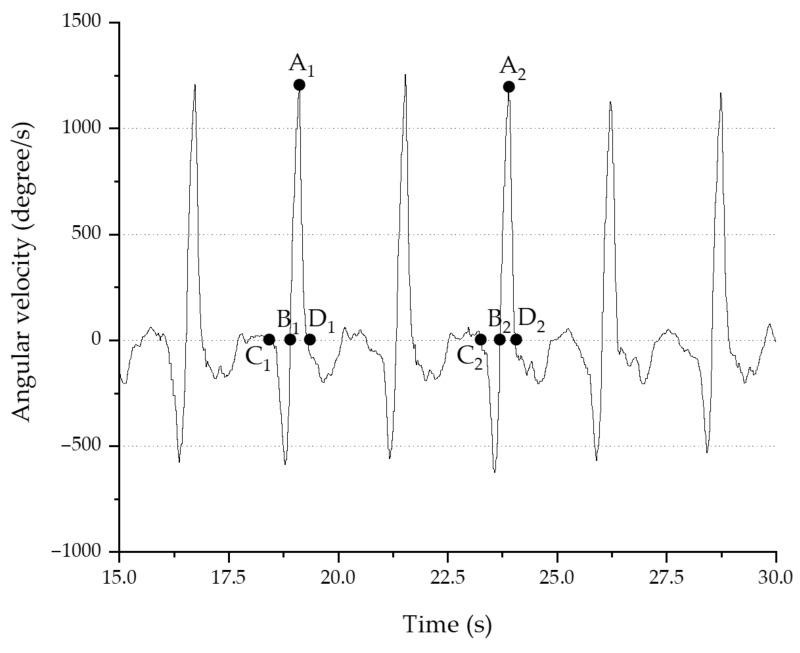
Forearm angular velocity measured from a gyroscope reading of the *z*-axis.

**Figure 3 sensors-21-03870-f003:**
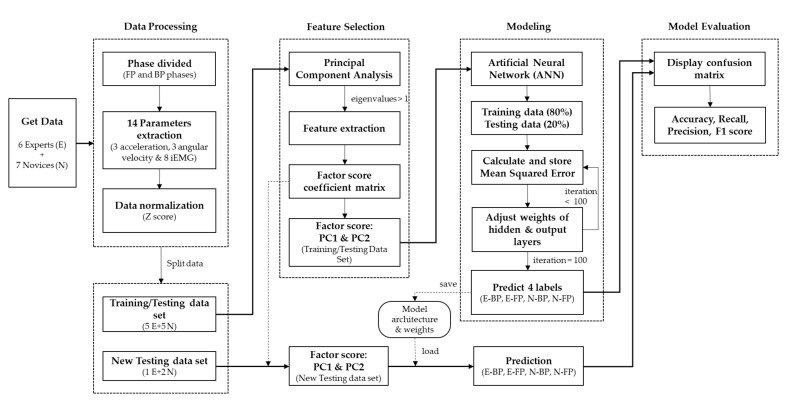
Flowchart of modeling steps.

**Figure 4 sensors-21-03870-f004:**
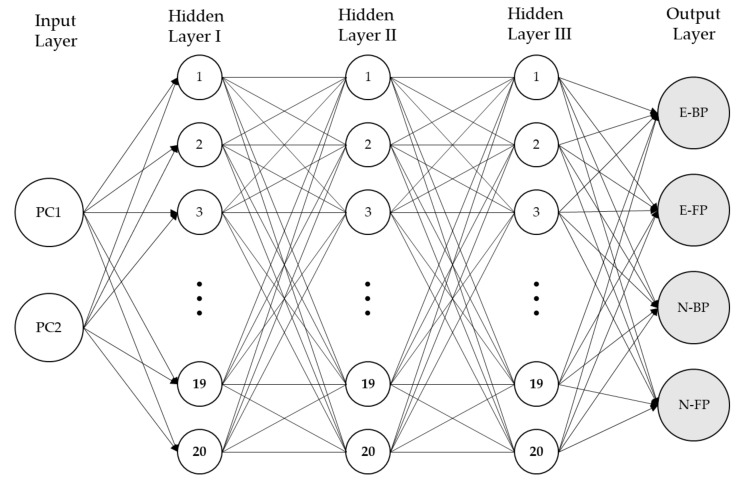
Artificial neural network (ANN) architecture.

**Figure 5 sensors-21-03870-f005:**
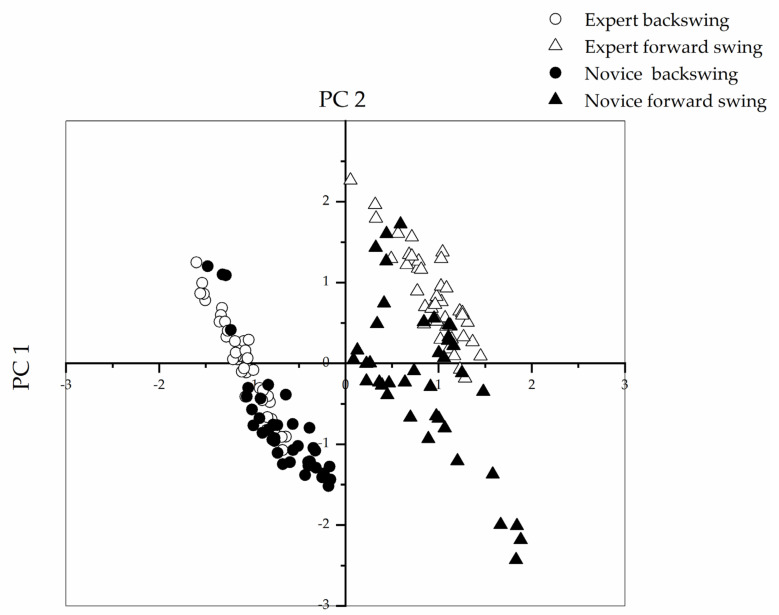
PCA loading plot of the first two principal components.

**Figure 6 sensors-21-03870-f006:**
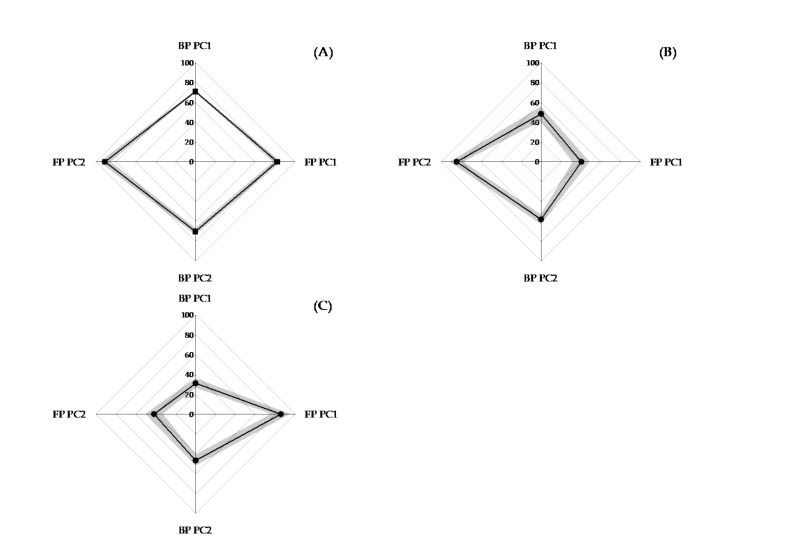
Radar graph for an expert and two novices (±1 SD gray shaded areas). (**A**) New expert and (**B**) new novice 1: sufficient wrist stability for paddle control, lack of attack power. (**C**) New novice 2: lack of grip strength and wrist stability for racquet control.

**Table 1 sensors-21-03870-t001:** Labeling of multiclass classification tasks.

Classification	Label Name
Expert backswing phase	E-BP
Expert forward swing phase	E-FP
Novice backswing phase	N-BP
Novice forward swing phase	N-FP

**Table 2 sensors-21-03870-t002:** Data for 14 features across two player levels during BP and FP phases (mean (SD)).

	E-BP	E-FP	N-BP	N-FP
Acc X	−1.51(0.38)	10.82(1.32)	−0.52(0.38)	5.56(3.02)
Acc Y	1.10(0.66)	−9.08(1.39)	0.93(0.57)	−5.93(2.45)
Acc Z	0.56(0.36)	−3.16(0.84)	0.37(0.29)	−2.46(1.18)
Gyro X	35.68(32.21)	−380.75(82.85)	56.38(45.48)	−319.67(200.27)
Gyro Y	136.97(49.42)	−465.58(136.51)	196.03(119.91)	−351.73(113.47)
Gyro Z	−352.28(137.00)	1192.91(69.11)	−489.94(226.25)	986.93(149.67)
EMG 1	10.73(4.34)	19.18(4.67)	4.87(1.97)	9.68(3.04)
EMG 2	15.81(3.89)	21.62(4.17)	9.80(5.32)	16.91(6.02)
EMG 3	10.99(3.35)	20.44(4.67)	9.82(4.16)	17.72(6.25)
EMG 4	8.77(2.29)	24.24(3.72)	10.47(5.10)	21.08(7.04)
EMG 5	8.87(3.40)	24.38(3.03)	9.18(5.17)	19.81(5.77)
EMG 6	6.35(3.23)	22.87(5.00)	6.89(5.74)	15.41(6.29)
EMG 7	6.61(1.81)	17.40(4.04)	6.65(5.88)	14.08(6.68)
EMG 8	10.23(3.61)	20.49(4.39)	5.97(5.51)	12.39(7.94)

Acc X, Y, Z, unit: g; Gyro X, Y, Z, unit: deg/s; EMG 1–8, unit: μV*s.

**Table 3 sensors-21-03870-t003:** Significant factors extracted by PCA.

	Forearm Explosive Force (PC1)	Wrist Muscle Control (PC2)
Gyro Y	−0.897	
Gyro Z	0.894	
Acc Y	−0.880	
Gyro X	−0.857	
Acc Z	−0.847	
Acc X	0.844	
EMG 2		0.886
EMG 7		0.796
EMG 8		0.792
EMG 1		0.787
EMG 3		0.722
EMG 4	0.613	0.684
EMG 5	0.644	0.653
EMG 6	0.604	0.647
Initial Eigenvalues	10.411	1.279
% of Variance	74.367	9.136
Cumulative %	74.367	83.503

Kaiser–Meyer–Olkin Measure of Sampling Adequacy = 0.896; Bartlett’s test of sphericity = 3589.490; degree of freedom = 91; significance < 0.01.

**Table 4 sensors-21-03870-t004:** Confusion matrix for the model testing and new testing data sets.

		Model Testing Data				New Testing Data			
	Predict	E-BP	E-FP	N-BP	N-FP	E-BP	E-FP	N-BP	N-FP
Label	E-BP	9	0	0	0	6	0	2	0
	E-FP	0	7	0	1	0	8	0	0
	N-BP	0	0	9	0	1	0	15	0
	N-FP	0	1	0	5	0	4	0	12

## Data Availability

The data presented in this study are available on request from the corresponding author.

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
