# Peer review of "Creating a Scoring System with an Armband Wearable Device for Table Tennis Forehand Loop Training: Combined Use of the Principal Component Analysis and Artificial Neural Network"

_sensors, 2021, doi:10.3390/s21113870_

Round 1

Reviewer 1 Report

This manuscript presented a table tennis scoring system enabled by the Myo armband for forehand loop training. 14 parameters were extracted in total for the analyses and machine learning procedures. Data from expert and novice has been collected for analysis. This proposed system has been demonstrated in classifying periods of table tennis strokes and can provide scores for performance quantification. In general, this proposed system shows a good performance catering to the future applications of training feedback and scoring of table tennis. Below are few minor comments:

  1. Are the sensing data collected through cables or transmitted to the nearby computer? If there are wires involved, will that affect the performance of the user or player?
  2. The recognition accuracy of new data dropped from 93.75% (testing) to 85.42%. Can the authors address this phenomenon?

Reviewer 2 Report

This well-written paper demonstrates the practical application of blending principal component analysis with ANN to produce insight on athletic training. The numerous steps involved in data collection can serve as a model for other research teams exploring kinetic/EMG athlete data. There are just a few minor points the authors may want to consider.

Lines 132-146, and then 158-166 really are the centerpiece of the work, and it would help clarify the processes to the reader with a flow chart.  This way, critical assumptions and setps can be identified.  For instance, why the specific cut-off frequency [line 143].  As well, for the ANN, neural networks have three types of layers: input, hidden, and output.  While 3 layers works well for most problems, what would be the basis for three hidden layers in this work? Indeed, N>1 for deep neural networks, but why the choice?  A flowchart will make more visible the structure of the analysis.

For the standard deviation, line 187, this seems to be a defined quantity (bracket) rather than a derived statistical quantity.  Is there another term or description that can convey that the thresholds were arbitrarily assigned?

The feature data of table 2 shows what might seem to be interesting results, but there is minimal interpretation of what it really means. are Gyro Y,Z values >100 significant? Comparison with a baseline?   Again, the jump from Table 2 to 3 might be facilitated with a process chart.

The emphasis on the process charts to elucidate the research effort will offset the weakness of the limited number o subjects used in the study.  Since there is no power law analysis to establish the minimum number of subjects required, it is thus important -- as the paper title suggests -- to ensure greater clarity on the process itself.

The paper affirms the model is satisfactory, but the authors do not reflect on weaknesses that might be present, and what the consider to be critical assumptions.  Indeed, there are just a few subjects involved and a LOT of data processing going on, so the journey from data to knowledge must have points of contention.

Reviewer 3 Report

This paper reported a table tennis skill recognition and classification system. The system uses wearable armband for angular velocity, acceleration, and electromyographic data collection, and Principal Components Analysis and neural network for data extraction and classification. The system can effectively recognize the table tennis skills of the players and quantify the player’s performance. The method developed in this research can potentially help the table tennis players to improve their skills through evidence-based and quantitative assessment. This paper is well written, and the study is clearly presented. I only have some minor comments as described below. This paper can be published in Sensors after minor revision.

  • How heavy is the Myo Armband? The armband looks bulky. Could the armband affect the performance of the players?
  • For the same player, how reproducible can this system recognize their performance?
  • It's not clear how this automatic assessment system can help the novices to improve their training efficiency. How to translate the score into a practical training approach?
